



# Comparing estimation techniques for timescale-dependent scaling of climate variability in paleoclimate time series

Raphaël Hébert [1], Kira Rehfeld [2], and Thomas Laepple [1,3]

[1]Alfred-Wegener-Institut Helmholtz-Zentrum für Polar- und Meeresforschung, Telegrafenberg A45, 14473 Potsdam, Germany
[2]Institut für Umweltphysik, Ruprecht-Karls-Universität Heidelberg, Im Neuenheimer Feld 229, 69120 Heidelberg, Germany
[3]University of Bremen, MARUM – Center for Marine Environmental Sciences and Faculty of Geosciences, 28334 Bremen, Germany

**Correspondence:** Raphaël Hébert, raphael.hebert@awi.de

**Abstract.** Characterizing the variability across timescales is important to understand the underlying dynamics of the Earth system. It remains challenging to do so from paleoclimate archives since they are more than often irregular and traditional methods to produce timescale-dependent estimates of variability such as the classical periodogram and the multitaper spectrum generally require regular time sampling. We have compared those traditional methods using interpolation with interpolation-free methods, namely the Lomb-Scargle periodogram and the first-order Haar structure function. The ability of those methods to produce timescale-dependent estimates of variability when applied to irregular data was evaluated in a comparative framework using surrogate paleo-proxy data generated with realistic sampling. The metric we chose to compare them is the scaling exponent, i.e. the linear slope in log-transformed coordinates, since it summarizes the behaviour of the variability across timescales. We found that for scaling estimates in irregular timeseries, the interpolation-free methods are to be preferred over the methods requiring interpolation as they allow for the utilization of the information from shorter timescale which are particularly affected by the irregularity. In addition, our results suggest that the Haar structure function is the safer choice of interpolation-free method since the Lomb-Scargle periodogram is unreliable when the underlying process generating the timeseries is not stationary. Given that we cannot know a priori what kind of scaling behaviour is contained in a paleoclimate timeseries, and that it is also possible that this changes as a function of timescale, it is a desirable characteristic for the method to handle both stationary and non-stationary cases alike.

## 1 Introduction

Paleoclimate archives are crucial to improve our understanding of climate variability on decadal to multi-centennial timescales and beyond (Braconnot et al., 2012). They provide independent data for the evaluation of the climate models which are used to project future climate change. To this end, quantitative estimates of variability based on palaeoclimate records allow to compare past changes in variance over a wide range of timescales (e.g. Laepple and Huybers, 2014b, a; Rehfeld et al., 2018; Zhu et al., 2019) and how variance is distributed across different timescales (e.g. Mitchell et al., 1976; Huybers and Curry, 2006; Lovejoy, 2015; Nilsen et al., 2016; Shao and Ditlevsen, 2016).





We generally refer to scaling as the statistical characterization of the changes in climate variability as a function of timescale $\tau$, or equivalently as a function of frequency $f$ such that $f = \tau^{-1}$. The term scaling often implies, although not necessarily,

power-law scaling: a stochastic process is said to be power-law scaling over a range of timescales $[\tau_1, \tau_2]$ if a time-scale dependent statistical metric $S(\tau)$ approximately follows a power-law relationship such that $S(\tau) \propto \tau^a$, where $a$ is a general power-law scaling exponent. Therefore, processes that fulfill this property show a log-linear relationship in the power spectrum (Schuster, 1898; Percival and Walden, 1993) over a given range of timescales. The corresponding power-law scaling exponent can be informative of the underlying dynamics of the system, such as the degree of temporal auto-correlation, i.e. the system's

memory (Mandelbrot and Wallis, 1968; Lovejoy et al., 2015; Graves et al., 2017; Fredriksen and Rypdal, 2015; Del Rio Amador and Lovejoy, 2019). While assuming power-law scaling may be a simplification, it is a rather accurate first-order description for a vast range of geophysical processes (Cannon and Mandelbrot, 1984; Pelletier and Turcotte, 1999; Malamud and Turcotte, 1999; Fedi, 2016; Corral and Gonz\'alez, 2019).

Methods used for scaling analysis generally assume that the process under investigation has been sampled at regular time

steps. This is appropriate for some instrumental observations and annually-resolved paleoclimate archives such as tree- and coral- rings. However, since most paleoclimate archives are the product of slow and intermittent accumulation in sediments or ice sheets, sampling them at regular depth intervals translates to irregular time intervals (Bradley, 2015). In addition, the irregular accumulation process usually has to be reconstructed and necessarily introduces age-uncertainty (Rehfeld and Kurths, 2014), although it does not affect the scaling estimates strongly (Rhines and Huybers, 2011).

Therefore, the primary challenge is that scaling analysis methods need to be adapted for the analysis of sparse and irregular series. Two approaches to this problem can be distinguished.

Firstly, an interpolation routine can be employed prior to the analysis in order to regularize the series. Once the series are regular, traditional methods such as the classical periodogram (CPG) or the multitaper spectrum method (MTM; Thomson, 1982) can be used (See this approach used in a paleoclimatology context in Huybers and Curry, 2006; Laepple and Huybers,

2014a, b; Rehfeld et al., 2018).

Secondly, the estimator can be adjusted for arbitrary sampling times. The so-called Lomb-Scargle Periodogram (LSP; Lomb, 1976; Scargle, 1982) was developed in the field of astronomy to identify periodic components in noisy astronomical timeseries with sampling hiatus and has sometimes been used in paleoclimatological context (Schulz and Stattegger, 1997; Trauth, 2020), although the LSP may introduce additional bias and variance (Schulz and Stattegger, 1997; Schulz and Mudelsee, 2002;

Rehfeld et al., 2011). More recently, Lovejoy and Schertzer (2012) advocated for the use of the Haar Structure Function (HSF), based on Haar wavelets (Haar, 1910), in geophysics due to its ease of interpretation and accuracy. Incidentally, the HSP can easily be adapted for irregular timeseries.

Another method which is often used for scaling analysis is the detrended fluctuations analysis (Peng et al., 1995; Kantelhardt et al., 2002, DFA) which has also been applied to climatic and paleoclimatic timeseries (Koscielny-Bunde et al., 1996; Rybski

et al., 2006; Shao and Ditlevsen, 2016). However, a pre-study showed it was less efficient for irregular timeseries and we decided to omit it for clarity. In addition, it underestimates variance at any given timescale because of the necessary detrending (Nilsen et al., 2016) and it is therefore of limited use beyond the estimation of scaling exponents.





In the present work, we compare the different methods for the scaling of variability and make them accessible in a single software package. Our main aim is to assess their ability to estimate variability across timescales in a paleoclimatological context which often entails scarce and irregular timeseries. In order to benchmark the methods, we evaluate them on surrogate data with known properties similar to those of palaeoclimate records without abrupt transitions. Finally, we apply the methods to a database comprising LGM and Holocene records to evaluate their performances on real data.

## 2   Data and Methods

In this section, we outline the different methods considered to evaluate scaling, and how they can be compared. We also provide a method to generate "paleo-proxy" surrogate data with realistic variability and sampling, which is then used to test and compare the scaling methods.

### 2.1   Scaling estimation methods

Scaling generally refers to the behaviour of a quantity $S(\tau)$ as a function of scale $\tau$ (or frequency $f$ such that $f = \tau^{-1}$) for a given process $X(t)$. In the current work, we exclusively consider timeseries, but the same methods can be used to investigate spatial scaling relationships. The quantity $S(\tau)$ considered can be the statistical moments of an appropriately defined fluctuation $\Delta X$ such as the power spectral density or Haar fluctuations. It is usual to assume such form to define a structure function for the statistical moments (Schertzer and Lovejoy, 1987; Lovejoy and Schertzer, 2012) of the process under investigation:

$$S_{\Delta X,q}(X,\tau) = <\Delta X_{j,\tau}^{q}> \propto \tau^{\xi(q)} \tag{1}$$

where "$< ... >$" denotes ensemble averaging over all fluctuations $j$ available at the given scale $\tau$, $q$ is the statistical moment (i.e. $q = 1$ corresponds to the mean, $q = 2$ to the variance, $q = 3$ to the skewness and so on), and $\xi(q) = qH - K(q)$ is the exponent function where $H$ is the fluctuation scaling exponent and $K(q)$ is the moment scaling function. $K(q)$ is zero for Gaussian processes and thus, for this specific case, all statistical moments scale similarly, i.e. $\xi(q) \propto qH$.

The equivalent metric for the power spectrum method is the spectral scaling exponent $\beta$ (see below section 2.1.1 for a formal definition). The two scaling exponents can be related by the following:

$$\beta = 1 + 2H - K(2) \tag{2}$$

where $K(q)$ at $q = 2$ is used since the spectrum is a second-order statistic. This relation can be understood intuitively since $\beta$ describes the scaling of the power spectral density obtained via the Fourier transform of the auto-covariance function whereas $H$ describes the scaling of the real space fluctuations. Therefore, since the auto-covariance is proportional to the expectation value of the (zero-mean) timeseries squared, the exponent $H$ is multiplied by two, while the integration in the definition of the Fourier transform increases the scaling exponent by $+1$.

In this work, we will focus on the quasi-Gaussian case in order to minimize the number of estimated parameters; paleoclimate archives often lack the resolution and/or length to estimate many parameters with confidence. This assumption is rather well





justified for temperature and precipitation timeseries at timescales longer than the turbulent weather regime (i.e. timescales longer than weeks or a few years depending on the location and medium (Lovejoy and Schertzer, 2012)). On the other hand,
highly intermittent archives which clearly display multifractality, such as volcanic series (Lovejoy and Varotsos, 2016) or dust flux (Lovejoy and Lambert, 2019), would require the "intermittency correction" from the moment scaling function $K(q)$.

Under the quasi-Gaussian assumption, we can simplify equation 2 since this implies $K(2) \approx 0$ and therefore:

$$\beta \approx 1 + 2H \qquad (3)$$

This allows to convert estimated $\hat{\beta}$ (where "ˆ" denotes an estimator for the given quantity) into their equivalent estimated
fluctuation scaling exponents $\hat{H}$. We use the fluctuation scaling exponent $H$ for inter-comparison between the methods in this study. $H$ is a natural choice as it describes directly the behaviour of fluctuations in real space and is the usual parameter in functions to generate fractional noises (Mandelbrot and Van Ness, 1968; Mandelbrot, 1971; Molz et al., 1997). While the fluctuation exponent $H$ takes its origin in the so-called Hurst exponent (Hurst, 1956), its meaning and definition has evolved and changed over time, see Graves et al. (2017) and Lovejoy et al. (2021) for historical summaries.

The process to estimate the scaling exponents can be divided into three steps. First, if the series is irregularly spaced, it requires to be regularized in order to be usable for the CPG and the MTM. The regularization is not necessary for the HSF and the LSP which have the advantage that they can be calculated directly on the irregular timeseries. Second, the fluctuations proper to each method are calculated as a function of timescale, and finally, the scaling exponent is fitted on the result.

### 2.1.1 Power Spectrum

For an ergodic, weakly stationary stochastic process $X$, the power spectrum $P$ is given by the Fourier transform of the auto-covariance function $\gamma$, and equivalently the squared Fourier transform of the signal (Storch and Zwiers, 1984),

$$P(\tau) = |\mathfrak{F}\{X\}|^2 = \mathfrak{F}(\gamma\{X\}). \qquad (4)$$

where $\tau$ is the timescale and $\mathfrak{F}$ denotes the Fourier transform operator. Equivalently, the power spectrum can be given as a function of the frequency $f = \tau^{-1}$ instead of the timescale $\tau$. We choose to write it as a function of timescale to allow for
visual comparison with the Haar method below, and also because it is more intuitive: a non-expert can easily grasp what the 1000-year timescale means rather than the equivalent 0.001-year$^{-1}$ frequency.

**Classical Periodogram**

The power spectrum for a discrete process $X$ with $N$ timesteps at regular intervals can be estimated using the classical periodogram (Chatfield, 2013):

$$\hat{P}_C(\tau) = \frac{1}{\pi N} \left| \sum_{t=1}^{N} X_t e^{\frac{-2\pi i t}{\tau}} \right|^2 \qquad (5)$$

If $\log(P(\tau))$ behaves linearly over a range of timescales $\tau$, the timeseries is considered to be power-law scaling over this timescale band, i.e. $P(\tau) \approx A\tau^\beta$ for an arbitrary constant $A$.





**Multitaper Spectrum**

The MTM method improves upon the CPG by producing independent estimates using a set of orthogonal functions: the prolate
spheroidal wave functions $h_{t,k}$ (Slepian and Pollak, 1961), also known as the Slepian tapers, which have the desirable property
of minimizing spectral leakage (Thomson, 1982). The spectral individual estimates for the $k^{th}$ taper can be written in a form
similar to the periodogram above:

$$\hat{P}_k(\tau) = \frac{1}{\pi N} \left| \sum_{t=1}^{N} h_{k,t} X_t e^{\frac{-2\pi it}{\tau}} \right|^2 \tag{6}$$

The estimator can then be expressed as the mean of the $K$ tapered estimates:

$$\hat{P}_{MT}(\tau) = \frac{1}{K} \sum_{k=0}^{K-1} \hat{P}_k(\tau) \tag{7}$$

**Interpolation**

To produce the power spectrum with the methods above, it is necessary to interpolate the series at a regular resolution. Follow-
ing Laepple and Huybers (2013), the data were first linearly interpolated to 10 times the mean resolution, then low-pass filtered
using a finite response filter with a cut-off frequency of 1.2 divided by the target mean resolution in order to avoid aliasing.
Linear interpolation corresponds to a convolution in the temporal domain with a triangular window. The Fourier transform of
the triangular window is $sinc^2$, where the $sinc$ function is defined as $sinc = x^{-1} sin(x)$, and therefore, a linear interpolation
multiplies the power spectrum by $sinc^4$ modulated by the resolution of the interpolation (Smith, 2011), resulting in a power
loss near the Nyquist frequency.

**Lomb-Scargle Periodogram**

As an alternative to the classical periodogram which requires regular data, Scargle (1982) introduced the LSP as a generalized
form of the CPG (equation 5):

$$\hat{P}_{LS}(\tau) = \frac{A^2}{2} \left[ \sum_j^N X(t_j) \cos \frac{2\pi(t_j - T)}{\tau} \right]^2 + \tag{8}$$

$$\frac{B^2}{2} \left[ \sum_j^N X(t_j) \sin \frac{2\pi(t_j - T)}{\tau} \right]^2 \tag{9}$$

where $A$, $B$ and $T$ are arbitrary functions of timescale $\tau$ and sampling times $t_j$ which can be irregular. We see that if $A =$
$B = \sqrt{\frac{2}{N}}$ and $T = 0$, we would recover the classical periodogram in case of regular sampling, but the reduction is not unique
and other choices of $A$ and $B$ can be made. The periodogram estimates are chi-squared distributed with 2 degrees of freedom





(i.e. an exponential distribution) and therefore, $A$ and $B$ will be chosen to retain this property. For independent and identically distributed white noise, it is the case when:

$$A(\tau, t_j) = \sqrt{\sum_j^N \cos^2 \frac{2\pi t_j}{\tau}} \tag{10}$$


$$B(\tau, t_j) = \sqrt{\sum_j^N \sin^2 \frac{2\pi \nu t_j}{\tau}} \tag{11}$$

The CPG is invariant to time translation since shifting the timesteps by a constant value only affects the phase of the complex exponential inside the absolute value. The function $T$ is thus introduced for the generalized periodogram to retain that property, which is the case when:

$$T(\tau, t_j) = \frac{\tau}{4\pi} \tan^{-1} \left( \frac{\sum_j^N \sin 4\pi \tau^{-1} t_j}{\sum_j^N \cos 4\pi \tau^{-1} t_j} \right) \tag{12}$$

The LSP has been mostly used in astronomy to detect periodic components in noisy irregular data. As outlined above, the functions $A$ and $B$ are chosen following the assumption that the process $X$ is approximately white noise with no temporal correlation. This assumption is of course problematic from the perspective of climate timeseries which usually exhibit long-range temporal correlation, but, as we will see later, good estimates of scaling exponents can nonetheless be recovered over a 155 wide range of $H$.

### 2.1.2 Haar Structure Function

The HSF allows to perform the scaling analysis in real space, i.e. without performing a Fourier transform. When used to define a structure function, Haar wavelets are appropriate to estimate the fluctuation exponent of processes with $H \in (-1, 1)$ (Lovejoy and Schertzer, 2012), a range covering most geophysical processes, and therefore, also because of its readiness to be applied 160 to irregular series and its ease of interpretation, Lovejoy and Schertzer (2012) argued it makes it a convenient choice for the scaling analysis of geophysical timeseries.

Haar fluctuations $\mathcal{H}$ are simple to implement for irregular sampling. They can be defined for a given interval of length $\tau$ as the difference between the mean of the first half of the interval with the second half:

$$\mathcal{H}_{\tau,j}(X(t_i)) = \frac{2}{\tau} \left| \sum_{t_j + \frac{\tau}{2} < t_i < t_j + \tau} X(t_i) - \sum_{t_j < t_i < t_j + \frac{\tau}{2}} X(t_i) \right| \tag{13}$$

The discrete sampling $t_i$ does not have to be regular since the average of the available fluctuations in the intervals is taken. This is of course an approximation since the fluctuations available might not exactly correspond to the expected timescale.

The first-order HSF $\mathcal{S}_{\mathcal{H}, q=1}$ can now be defined by using the Haar fluctuation defined above in equation 1 and letting $q = 1$:

$$\mathcal{S}_{\mathcal{H}, q=1}(X(t), \tau) = < \mathcal{H}_{\tau,j}(X(t)) > \approx A\tau^H \tag{14}$$





Higher-order structure functions can be defined by considering other values of $q$ and analyze multifractal processes, but since we are only dealing with the quasi-Gaussian case in this manuscript, $q = 1$ is sufficient. For simplicity, we refer to the first-order HSF simply as HSF.

Since we are taking the average value of the absolute of the fluctuations, if the fluctuations are distributed in a quasi-Gaussian fashion, then we expect the distribution of Haar fluctuations at each scale to be a half-normal distribution, i.e. a zero-mean Gaussian distribution folded along zero. The mean $\mu_{HG}$ of such half-Gaussian distribution, which can be obtained by taking the absolute of a Gaussian distribution, is proportional to the standard deviation $\sigma_G$ of the original Gaussian distribution, i.e. $\mu_{HG} = \sqrt{\frac{2}{\pi}} \sigma_G$. Since the sign of the Haar fluctuation before taking the absolute is arbitrary, we can also consider the negative of those fluctuations to create an ensemble which is even closer to a Gaussian distribution and has mean equal zero by construction. We then estimate the mean absolute Haar fluctuation using the definition with the standard deviation given above. This procedure marginally improves the estimation procedure over directly taking the average of the absolute of the Haar fluctuations available.

### 2.1.3 Slope Estimation

For a power-law relationship between variables $x$ and $y$ such as $y = Ax^B$, it is usual to use standard least-square fitting to find the coefficient $A$ and the exponent $B$ as the linear coefficients of the equivalent linear relationship after taking the logarithm of the equation such that: $\log y = B \log x + \log A$. Least-square fitting assumes the residuals are normally distributed which is often a good approximation for the logarithm of the power spectral density of a stochastic process at a given frequency.

In the case of stationary Gaussian processes, it can be shown that the CPG and LSP estimates at a given timescale are distributed like a chi-square distribution with degrees of freedom equal to two (i.e. an exponential distribution), and for the MTM it is approximately twice the number of tapers, albeit slightly less depending on their degree of dependence. The logarithm of the distributions mentioned above is similar enough to a normal distribution to obtain reasonable estimates with an ordinary least-square fit. Another option is to use a generalized linear model with a gamma distribution model which the chi-square is a special case of. While very similar results are obtained for both fitting methods, we chose the generalized linear model method since it is theoretically more justified.

For a Gaussian process, the first-order Haar fluctuations will follow a normal distribution, but as we are fitting the absolute value of the fluctuations, the estimates at a given timescale will follow a half-normal distribution as mentionned above. Although the half-normal distribution is not a specific case of the gamma distribution, but rather the generalized gamma distribution, we also use the generalized linear model to estimate its scaling exponent for practical purposes.

**Timescale Band for Slope Estimation**

For all methods, we need to select a range of timescales over which to estimate the scaling exponent using the generalized linear model. The maximum fitting timescale $\tau_{max}$ used was always one third of the largest scale available in order to maximize the range of timescales used while avoiding the longer timescales which have poor statistics in the case of the HSF (Lovejoy and Schertzer, 2012), and underestimate variance in the case of the MTM because of its known bias (Prieto et al., 2007) and the





usual linear detrending. Since the impact of irregularity is not important at long timescales, we chose this same $\tau_{max}$ for all methods for consistency. The optimal minimum timescale $\tau_{min}$ on the other hand can vary depending on the method used and requires a careful treatment.

For the CPG and the MTM, we fitted above the timescale corresponding to the Nyquist frequency, i.e twice the resolution, for regular series. In the case of irregular data, since the interpolation brings about a power loss at small timescales (Schulz and Stattegger, 1997; Rhines and Huybers, 2011; Kunz et al., 2020), fitting over the small timescales produces a positive bias on the slope estimation. Therefore, when estimating the scaling exponents for the irregular series, we consistently fitted above three times the mean resolution $\tau_\mu$ as a compromise (i.e. 1.5 times the Nyquist frequency). This choice of $\tau_{min}$ was informed

(as for the methods below) from the results using the paleoclimate database (see section 3.3).

For both the HSF ans LSP, we used twice the resolution as $\tau_{min}$ for the regular cases and we used $\tau_{min} = 2\tau_\mu$ for the irregular cases.

## 2.2    Evaluation of the estimators

### 2.2.1    Surrogate data

To test the methods, we produce surrogate data with the same characteristic as the paleoclimate archives, namely a given scaling behaviour and an irregular sampling in time.

**Simulation of power-law noise**

A classical example of a non-stationary process ($H > 0$) is normal Brownian motion ($H = \frac{1}{2}$) which is produced by (integer order) integration of a normal Gaussian white noise ($H = -\frac{1}{2}$). A process with a given scaling behaviour can be obtained

by fractional, rather than integer, order integration (or differentiation) of a given set of innovations, i.e. a random series of uncorrelated values obtained from a certain distribution. To generate our surrogate data, we consider the simplest case when the innovations are drawn from a normal Gaussian distribution. This leads to the classical fractional Gaussian noise (fGn) and fractional Brownian motion (fBm) (Mandelbrot and Van Ness, 1968; Mandelbrot, 1971; Molz et al., 1997). For any fGn process there is a related fBm process which can be obtained by (integer-order) integration, which increases the scaling exponent of

the process by $+1$. It is usual to define the associated fGn and fBm processes by the same scaling exponent $h \in (0,1)$ which directly describes the scaling behaviour of the HSF for the fBm, or for the fGn after (integer order) integration. However, it is inconvenient to use the same parameter for both fGn and fBm when considering them in a common framework and we prefer to also identify the fGn by the scaling behaviour of its HSF, rather than that of its integral, such that it has $H \in (-1,0)$. In the current paper, we generally refer to both fGn and fBm as "fractional noise" described by an exponent $H \in (-1,1)$. They are

generated using an algorithm developed by Franzke et al. (2012).

Series with $H \in (-0.5,-0.3)$ are typical of monthly land air-surface temperature up to decadal timescales, while series with $H \in (-0.3,0)$ are more typical of sea-surface temperature over similar timescales. Non-stationary behaviour with $H > 0$ is typically observed in pre-Holocene series comprising Dansgaard-Oeschger events (Nilsen et al., 2016).





**Generation of irregular paleoseries**

To produce irregularly sampled series akin to paleoclimate archives, an "annual resolution" series with a given scaling exponent is first produced with the above method, and then degraded at the desired resolution using two different methods. The first one is to simply block average, and the second one is to sub-sample a low-pass filtered version of the series.

For the block averaging method, boundaries are determined as the midpoint between subsequent timesteps, and all data in between is averaged. This corresponds, in the temporal domain, to a convolution with a square window, or equivalently in 240 the frequency domain, multiplying the Fourier transform by the $sinc$ function ($sinc(x) = x^{-1}sin(x)$). Therefore, the power spectrum is multiplied by a $sinc^2$ and this brings about a loss of power at the high frequencies. For the second method, the timescale of the low-pass filter is taken as twice the mean resolution of the series, and the filtered series is simply sub-sampled at the desired timesteps. In the frequency domain, the filtering corresponds to multiplying the spectrum by a square function which cuts off variability below the specified timescale, and therefore, this is useful to reduce the aliasing of higher-frequencies.

The first method would correspond to an archive sampled without gaps and where there is no smoothing of the signal, for example speleothem and varved sediments if samples containing several layers are taken, or marine records when the sample distance is smaller than the typical mixing depth in the sediment (Berger and Heath, 1968). The second method would correspond to archives with spaced out sampling (for example 1 cm every 10 cm) and including processes such as bio-turbation and diffusion which smooth out the high-frequency signal, for example biochemical and ecological data extracted from sediment 250 cores (Dolman and Laepple, 2018; Dolman et al., 2020). We show the second method as our main result since it is more common to have archives with such smoothing processes, but also because it removes most aliasing effect from our results. It therefore leaves a clearer picture of the other effects inherent to each methods. It is however an idealization since the smoothing timescale of the physical processes involved is not related to the resolution; it should be independently estimated and accounted for in applied studies, although it is seldom reported. All the results were also computed for the block average method and are 255 shown in the supplementary information (Fig. S1, Fig. S2, Fig. S3, Fig. S4, Fig. S5, Fig. S6).

Previous studies have shown that the distribution of sampling times for typical sedimentary records can be approximated by a gamma distribution (Reschke et al., 2019a). To test systematically the impact of increasingly irregular sampling, we thus draw the time steps from a gamma distribution defined by its shape parameter $k$ (or skewness $\nu$ such that $k = \nu^{-1}$) and mean parameter $\mu$ which corresponds to the mean time-step $\tau_\mu$. When the skewness parameter is $\nu = 0$, then we have regular 260 sampling of width $\mu$. Figure 1 shows an example of such surrogate series at annual resolution and an irregularly degraded version (with timesteps drawn from a gamma distribution with $\nu = 1$), along with the results of applying the three scaling estimation methods considered for the main analysis: the MTM, the HSF and the LSP. We omit a discussion of the results computed using the CPG as they are generally very similar to those of the MTM, albeit with higher variance.

**2.2.2 Performance metrics and performance plots**

Our aim is to evaluate how the different methods perform in the estimation of scaling exponents for irregularly sampled paleoclimate data $X(t)$ with different scaling behaviour, and of variable length and irregularity. In order to assess the accuracy

**Figure 1. (a,e,i,m,q,u)** Surrogate timeseries generated with a given $H$ are shown at annual resolution (brown), degraded to a regular 50-year resolution (blue) and degraded to an irregular and random spacing drawn out of a gamma distribution with skewness $\nu = 1$ and mean 50 years. **(b,f,j,n,r,v)** Shown are the mean power spectra, estimated using the MTM, of 100 realizations of surrogate timeseries generated as in (a,e,i,m,q,u), respectively, and shown with the same colour scheme. The irregular case is also shown after dividing the power spectra by the expected $sinc^4$ bias due to interpolation (dashed pink). Also shown are the bounds for the fitting range considered later (vertical dashed blue): at the Nyquist frequency corresponding to 100 years, at 1.5 times the Nyquist frequency corresponding to 150 years and at one third the length of the timeseries at 2000 years. **c,g,k,o,s,w** Same as (b,f,j,n,r,v), respectively, but for the LSP instead of the MTM. **d**. Same as (b,f,j,n,r,v), respectively, but showing HSF instead of MTM power spectra.





and precision of our scaling exponent estimator $\hat{H}$ for a given set of parameters, we generate an ensemble of surrogate data $\hat{X}(t)$ and analyze its statistics. We evaluate three measures: the bias $B$ for the accuracy of the estimator $\hat{H}$ with respect to the input $H$, the standard deviation $\sigma$ for the precision of the estimator, and the root-mean-square error $RMSE$ which combines both previous measures:

$$B = < \hat{H} > - H \tag{15}$$

$$\sigma = \sqrt{< \hat{H}^2 > - < \hat{H} >^2} \tag{16}$$

$$\text{RMSE}^2 = B^2 + \sigma^2 \tag{17}$$

We exploited the geometric relation between the three to easily visualize the results. We summarize the results for a set of parameters by one point on a bias-standard deviation diagram (Fig. 3) where the x-axis gives the bias, the y-axis the standard deviation and the distance from the origin then corresponds to the root-mean-square error. The bias and standard deviation at for each combination is calculated from a large ensemble of surrogate data (10 000 realizations).

## 2.3 Data

In order to test the methods with the sampling properties of typical paleo-proxy data, we consider an available database of marine and terrestrial proxy records for temperature (Rehfeld et al., 2018), which was also used for signal-to-noise ratio assessments by Reschke et al. (2019b). Each of the 99 sites covers at least 4000 years in the interval of the Holocene (8–0 kyr ago, 88 time series in total) and/or the LGM (27–19 kyr ago, 39 time series total) at a mean sampling interval of 225 years or lower. These records are irregularly sampled in time, and come with different sampling characteristics (Fig. 2).

## 3 Results

Using the methods described above to generate synthetic data, we can evaluate the ability of each method to recover the input scaling exponents. For each method, we calculate the bias and standard deviation over the ensembles of surrogate data, for input $H$ between -0.9 and 0.9 in increments of 0.1. This wide range of $H$ values covers all values encountered later in the multi-proxy database; the vast majority of climatic series even fall within an even more restricted range of $H \in [-0.5, 0.5]$. First, we consider the ideal case of regular sampling, then we study the effect of irregular sampling and finally, the impact of the length of the time series.

### 3.1 Effect of Regular and Irregular Sampling

We evaluate the estimators for four cases pertaining to the resolution of the data and always a fixed length of 128 data points (Fig. 3). The first case considers "annual data" which was directly simulated and not degraded afterwards; it is therefore regular.


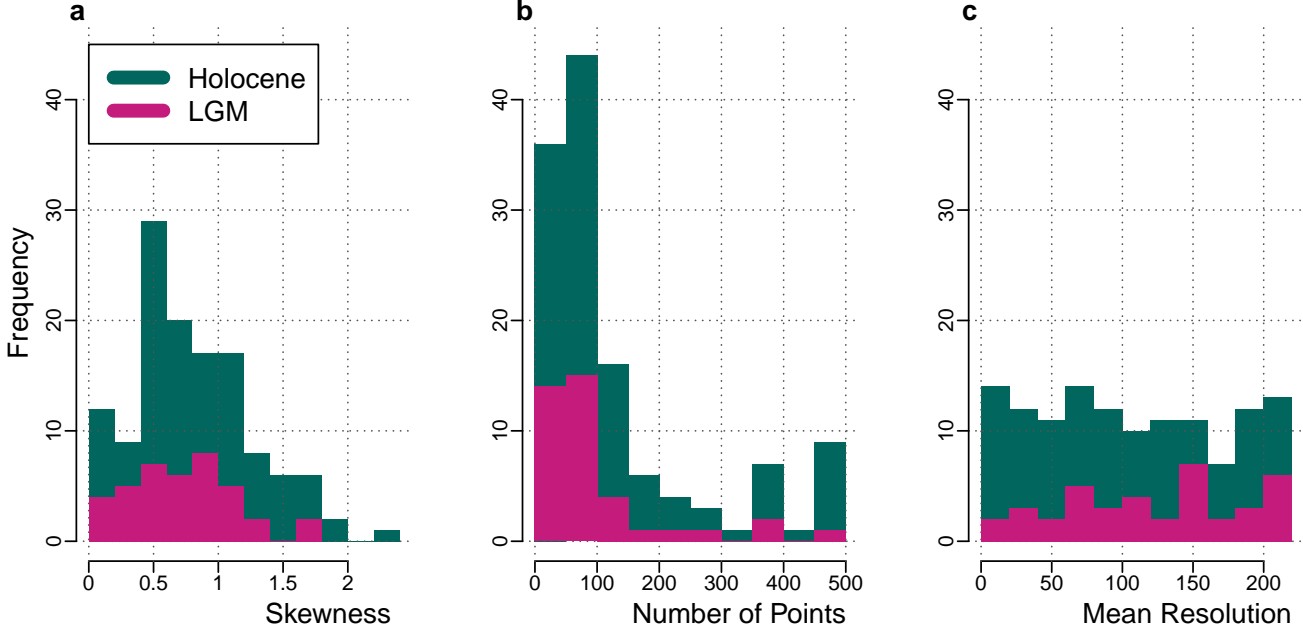

**Figure 2.** Sampling characteristics of the 127 paleoclimate timeseries (Rehfeld et al., 2018). The 88 Holocene series (teal) and 39 Last Glacial Maximum series (pink) are evaluated along the following characteristics: **(a)** the skewness $\nu$ of the distribution of time steps , **(b)** the number of points, i.e. samples , and **(c)** the mean resolution $\tau_\mu$ of the time steps.

It is shown for comparison with the second case where we simulate 5120-year long series and then degrade them to 40-year resolution. This allows us to study the impact of the degrading method which is necessary for producing irregular series. The third and fourth cases deal with series of 128 irregular timesteps drawn from a gamma distribution with skewness $\nu = 0.5$ and $\nu = 1$ respectively, and a mean parameter of 40-year so that the resulting series have the same mean resolution as the second

300 (regular) case. To illustrate the contribution of different frequency ranges to the precision and accuracy of the estimators, the scaling exponents were also fit on sub-ranges of equal width in the log of the timescales which we refer to as the shorter timescales, the intermediate timescales and the longer timescales, corresponding to, respectively, 2-9.2 $\tau_\mu$, 4.3-19.9 $\tau_\mu$ and 9.2-42.7 $\tau_\mu$, where $\tau_\mu$ is the mean resolution of the timeseries (Figure 4).

For the "annual data" series, the scaling exponent $H$ can generally be recovered for all values of $H \in [-0.9, 0.9]$ with a
305 standard devation below 0.13 and an absolute bias less than 0.06 (Fig. 3, top left), except for the LSP when $H > -0.1$ as it increasingly underestimates higher values of $H$. The small bias observed in the MTM and LSP estimates stems from the shorter timescales (Fig. 4) which are sensitive to aliasing of the power below the Nyquist frequency, and therefore the measured scaling slope of series with negative $\beta$ (i.e. $H < -0.5$) are biased high whereas increasingly negative bias is obtained for increasingly positive values of $\beta$ ($H > -0.5$). It follows that the flat case of $\beta = 0$ ($H = -0.5$) is unbiased by aliasing since the aliased


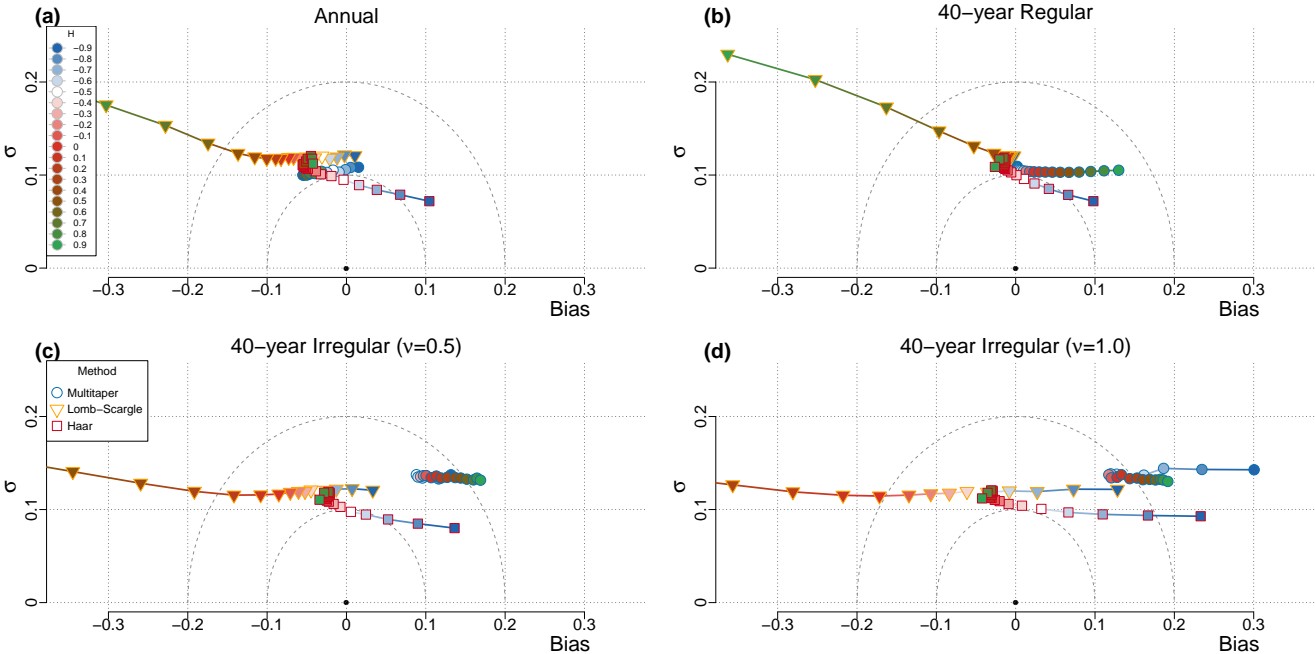

**Figure 3.** Bias-Standard Deviation diagram for ensembles of $H$ estimates surrogate timeseries with input values of $H \in (-1, 1)$ (i.e. $\beta \in (-1, 3)$). Different types of data are evaluated: **(a)** regular "annual data", i.e. it was directly simulated and not degraded after, **(b)** regular 'surrogate data degraded at regular 40-year interval, and irregular surrogate data with timesteps drawn from a gamma distribution with **(c)** skewness $\nu = 0.5$ or **(d)** skewness $\nu = 1$.

power is likewise flat. The HSF based estimates also have a significant bias in the other direction when the series considered have an input $H$ decreasing below $H = -0.5$.

The second case of regular 40-year series yields similar results as the previous case, although there is no more aliasing since we sub-sampled the 5120-year long series at 40-year intervals after an 80-year low-pass filter was applied (Fig. 3, top right). The MTM returns a consistent estimate with a variance which remains almost constant over the range of $H$. The estimator has little bias for stationary series with $H < 0$, but it steadily increases for higher values as the lower-frequencies bend upward (Fig. 4). This seems to be related to the bias low characteristic of the MTM at the longest timescales (Prieto et al., 2007) which creates an inflection point (around $\Delta t \approx 3000 \, \text{years}$ on Fig. 1) and the power lost to its right appears redistributed to its left. Therefore, as $H$ increases, the amount of power lost is more important and the inflection stronger. The LSP estimates are also largely unbiased until about $H = 0.5$, and above it a strong negative bias is developed, particularly on the side of the smaller timescales (Fig. 4, Fig. 1).

The problem of irregularity is considered using irregular surrogate data skewness parameter $\nu = 0.5$ and $\nu = 1$ (Fig. 3, bottom row). In the case of the MTM, we observe a consistent bias for all $H$ of about 0.5 and 0.1 for the shorter and intermediate timescales, respectively, for the weakly irregular case ($\nu = 0.5$), and practically no bias ( 0.02) at the longer timescales. This





**Figure 4.** Timescale dependence of the bias and variance for regular and irregular series. We evaluate the three methods: MTM (circles), HSF (square), and LSP (triangles). The colors corresponds to the input $H$ value for each simulation, ranging from -0.9 to 0.9. in increments of 0.1. The rows correspond to different types of surrogate series: **(a-c)** "annual data", **(d-f)** regular data , **(g-i)** mildy irregular data ($\nu = 0.5$), **(j-l)** strongly irregular data ($\nu = 1.0$); see section 2.2.1. The columns correspond to three different fitting ranges in terms of the mean resolution $\tau_\mu$, **(a,d,g,j)** the shorter timescales: 2-9.2 $\tau_\mu$, **(b,e,h,k)** the intermediate timescales: 4.3-19.9 $\tau_\mu$ and **(c,f,i,l)** the longer timescales: 9.2-42.7 $\tau_\mu$.





corresponds very closely to the expected bias from the linear interpolation step, and it could potentially be accounted and
corrected for. While it is outside of the scope of this study to formally develop such method, if we apply a first-order correction
by dividing the MTM spectra by a $sinc^4$ with time constant corresponding to the mean sampling resolution, we can greatly
reduce the bias down to the Nyquist frequency (Fig. 1, Fig. S7).

The negative bias for the LSP evoked above for $H > 0.5$ amplifies with irregularity and even affects significantly a wider
range of $H$ values, down to $H = -0.5$ for the most irregular case ($\nu = 1.$). This is a consequence of higher than expected
variance over the smaller timescales in the LSP when the slope is steep, as already identified by Schulz and Mudelsee (2002)
using red noise. The HSF estimates on the other hand remain mostly insensitive to the irregular sampling for all $H > -0.5$, but
the positive bias we already identified in the regular case when $H < -0.5$ amplifies for more negative values of $H$. Similarly, all
methods yield increasing positive bias for such anti-persistent timeseries (i.e. with $H < -0.5$) when the sampling is irregular.
This conjuncture seems to indicate that the anti-correlation characteristic of $H < -0.5$ processes is lost to some extent when
degraded in an irregular manner to produce the surrogate data, as they all show a similar increase of their bias (by about 0.15
for the $H = -0.9$ case) with respect to their bias for the $H = -0.5$ case.

### 3.2 Effect of Time series length

While the irregularity had a larger impact on the bias of the estimator, the length of the time series mostly influences the
variance of the estimator (Fig. S8).

In the case of the MTM estimates, all values of $H$ result in a similar standard deviation for a given resolution. When the
resolution is doubled, the standard deviation increases by a factor close to the expected $\sqrt{2}$ (by ~1.4-1.7), going from $\sigma \approx 0.27$
at the 160-year resolution to $\sigma \approx 0.07$ at the 20-year resolution (Fig. S8). The bias also improves when the resolution increases,
particularly for the higher values of $H$ such as for $H = 0.9$ which goes from $B \approx 0.27$ at the 160-year resolution to $B \approx 0.10$ at
the 20-year resolution, while for negative $H$ values it goes from $B \approx 0.05 - 0.07$ to $|B| < 0.01$ for the same resolution change.

In the case of the LSP, up to $H \approx 0.1$, the standard deviation of the estimates also decrease by a factor close to $\sqrt{2}$ (by
~1.4-1.8, Fig. S8) with each doubling of resolution, going from $\sigma \approx 0.35$ at the 160-year resolution to $\sigma \approx 0.08$ at the 20-year
resolution, and the bias improves only slightly since it was already small ($|B| < 0.04$). For the higher $H$ values, the standard
deviation improves less (by ~1.2-1.4 at $H = 0.9$, Fig. S8), going from $\sigma \approx 0.43$ at the 160-year resolution to only $\sigma \approx 0.19$ at
the 20-year resolution. At the same time, the bias change becomes more positive, thus compensating slightly better the overall
strong negative bias characteristic of the LSP estimates for very high $H$ values.

In the case of the HSF, when increasing the resolution, the standard deviation of the estimates improves more for the lower
$H$ values than for the higher values, improving by a factor of ~1.4-1.8 for $H = -0.9$ to ~1.2-1.4 for $H = 0.9$ (Fig. S8).
Overall, the standard deviation improves from $\sigma \approx 0.20 - 0.25$ at the 160-year resolution to $\sigma \approx 0.06 - 0.09$ at the 20-year
resolution. The slight negative bias found for the $H > -0.5$ estimates ($|B| < 0.04$) practically vanishes, but not the positive
one for $H < -0.5$ which only decreases slightly.


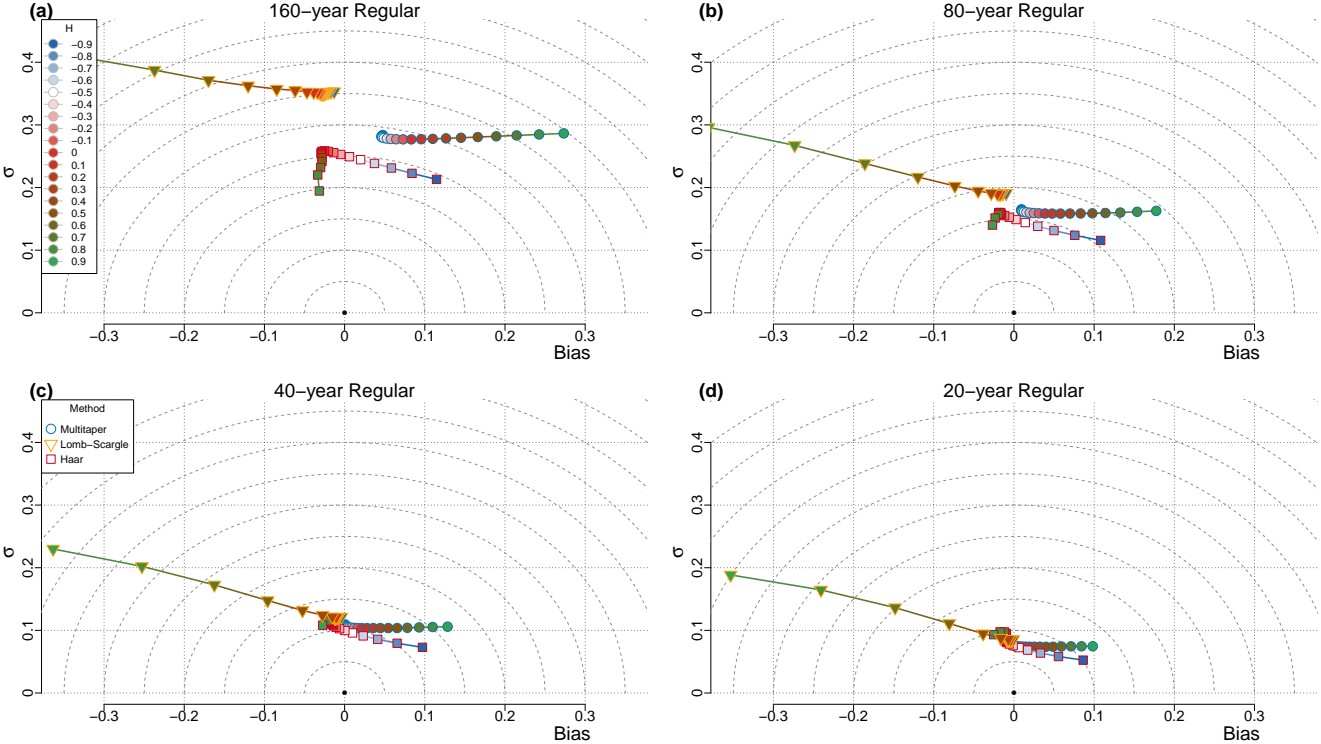

**Figure 5.** Resolution dependence of the estimators. The performance of the estimators is shown for regular surrogate data of different resolutions: **(a)** 160 years , **(b)** 80 years, **(c)** 40 years and **(d)** 20 years. For each case, the series spanned 5120 years, and therefore each case contains 32, 64, 128 and 256 data points, respectively.

### 3.3 Application to database

In order to see how these results translate to typical proxy records, we perform the analysis with surrogate data with sampling characteristic and scaling behaviour directly extracted from the database of Holocene and LGM records. We make the assumption that the series are approximately power-law scaling and that they can therefore be modelled by fractional noise described

by a scaling exponent $H$. Since the best $H$ to describe the approximate scaling behaviour of the series is unknown, we make an initial approximation with each method and take the median of the three results as the reference value for the given series, with which we generate an ensemble of surrogate with the same sampling scheme as the given series. We use the real timesteps in order to evaluate the impact of each specific sampling scheme. Again, we consistently fitted up to one third of the length (see section 2.1.3), but determined empirically the best minimum fitting scale $\tau_{min}$ such that it minimizes the RMSE of the

estimator with respect to the reference $H$ used to generate the surrogate data.

We found the HSF to yield the smallest $\tau_{min}$ of the three methods, with $\tau_{min}$ below twice the mean resolution $\tau_{\mu}$ for 121 out of 127 series and even below $\tau_{\mu}$ for 43 out of 127. The LSP yielded similar results with 109 out of 127 series having $\tau_{min}$


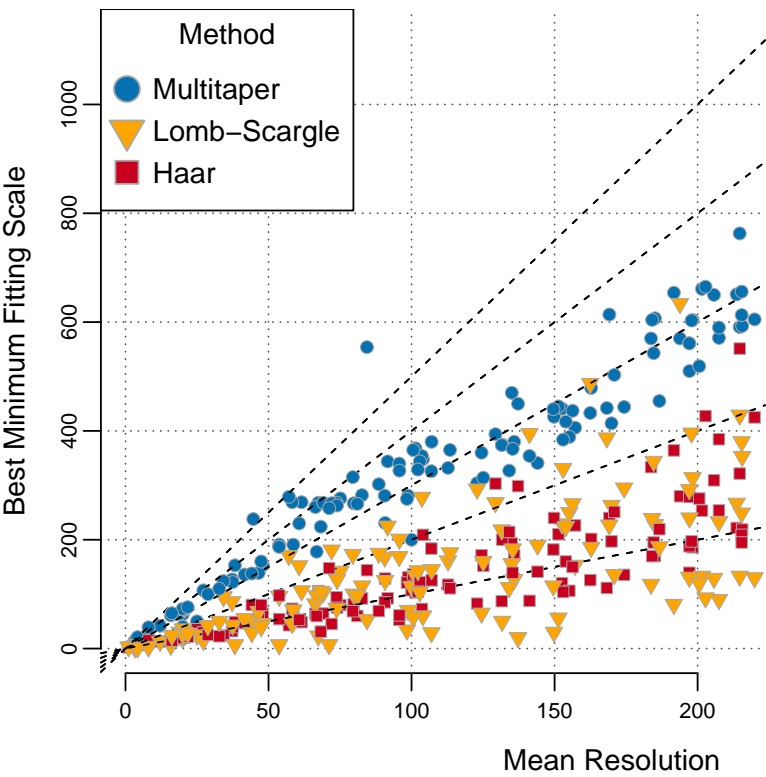

**Figure 6.** Best minimum fitting timescales. For each methods are shown the best minimum fitting timescales $\tau_{min}$ (minimizing the RMSE in $H$) as a function of the mean resolution $\tau_\mu$ for ensembles of surrogate data generated with the same sampling scheme as the corresponding paleoclimate timeseries from the database.

below twice $\tau_\mu$, and 41 out of 127 having $\tau_{min}$ even below $\tau_\mu$. This contrasts with the MTM which almost never suggests best results for $\tau_{min}$ below twice $\tau_\mu$, but rather an average minimum resolution $\bar{\tau}_{min} = 3.3\tau_\mu$, compared to $\bar{\tau}_{min} = 1.2\tau_\mu$ for the

HSF and $\bar{\tau}_{min} = 1.3\tau_\mu$ for the LSP. This underlines because interpolation-free methods give more reliable estimates at shorter timescales, they allow a better usage of the full data.

    The HSF yields the best results of all three methods both in terms of variance of the estimates, and in terms of bias, with a mild tendency towards positive biases. The MTM also tends to show positive biases, albeit higher, while the LSP tends to show negative biases. Both the MTM and the LSP generally show higher variance of their estimators than the HSF (Fig. 7), although

for different reasons. The MTM has higher variance because the higher $\tau_{min}$ does not allow to use the smaller timescales in the estimation procedure, while the LSP shows higher variance because of the timeseries with high input $H$ for which it performs poorly, especially when the data are irregular (Fig. 3).

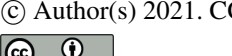


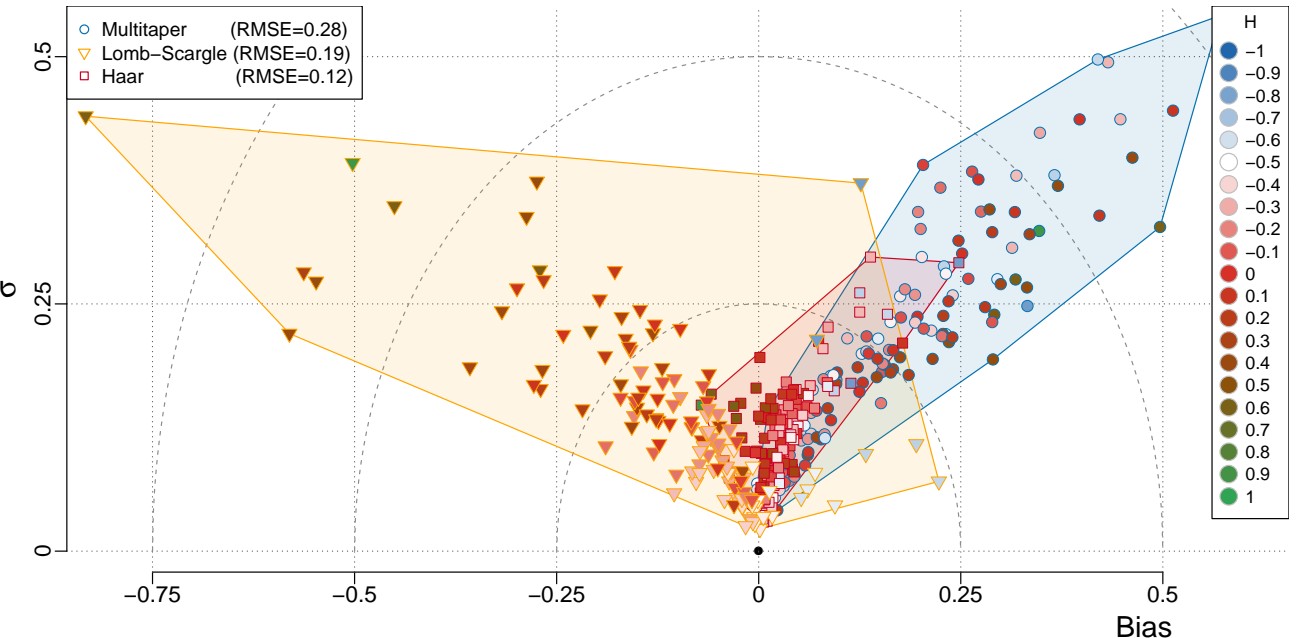

**Figure 7.** Bias-standard deviation diagram for the surrogate timeseries generated using timesteps from the paleoclimate database. The input $H$ used to generate the timeseries is indicated by the colour inside the markers. The shaded polygons contain all the points for a given method (see legend for colours).

On average, the HSF method gave the lowest $RMSE = 0.11 \pm 0.04$ compared to $RMSE = 0.30 \pm 0.15$ and $RMSE = 0.19 \pm 0.15$ for the MTM and the LSP, respectively. The poor performance of the LSP compared to the HSP stems from the higher $H$
since it performs as well as the Haar when $H < 0$, with $RMSE = 0.12 \pm 0.07$ (Fig. 8).

## 4   Discussion

Our comparative study indicates that for irregular timeseries the methods with interpolation, i.e.the CPG and the MTM, are less efficient to evaluate variability across timescales than the methods without interpolation, i.e. the LSP and the HSF. In the case of regular timeseries however, all methods were found to perform similarly (Fig. 3b) for a wide range of input $H$,
and only showed bias on the fringes of the $H$ range considered. In addition, we found that the choice of method should also be informed by the characteristics of the underlying process measured since there was an observed dependence between the performance metrics and the scaling exponent $H$ which generated the surrogate data. As such, the LSP may be only appropriate for irregular series suspected of having $H$ near or below $H = -0.5$, while the HSF shows better reliability even when $H$ is near or above $H = 0$. It might not be so surprising that the LSP performed poorly for higher $H$ values since it has been developed
to approximate the CPG for stationary noise processes (Scargle, 1982) and processes with $H > 0$ are non-stationary, i.e. their mean is not stable and changes with time.


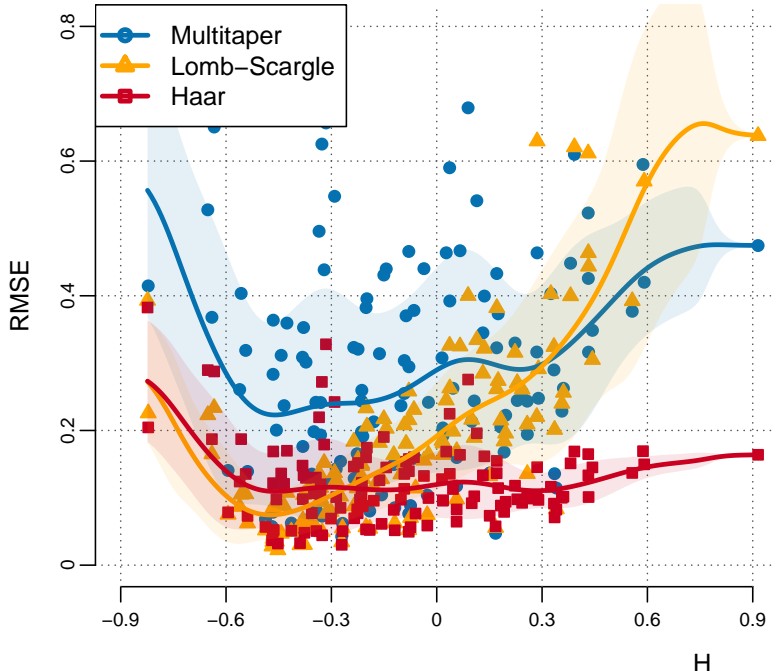

**Figure 8.** RMSE of the $H$ estimates as a function of the input $H$ for the surrogate timeseries based on the proxy database. The RMSE are given for the three estimation methods as a function of the $H$ used to generate the surrogates. Also shown is a Gaussian smoothing of the points for each method (thick line) and the one standard deviation confidence interval (shaded).

In addition to the irregularity, we also compared how the resolution of the timeseries affects the estimators of $H$. While making the series more irregular mostly affected the accuracy of the estimates, i.e. their bias, without affecting their precision, i.e. their standard deviation, increasing the resolution mostly improved the precision and the accuracy only to a lesser extent. Increasing the resolution naturally improves the estimates as it provides more information and also because it decreases the relative weight of the most uncertain timescales: the smaller timescales, near the sampling resolution, and the longer timescales, near the length of the series.

Our conclusions however rely on the real data having similar properties to the surrogate data used to validate the methods, and the conclusions may be different if the real data have properties different than the assumed power-law scaling. It is difficult to say for example which method would perform best if the data analyzed would contain two different scaling regimes, one with low $H$ and the other with high $H$ (Lovejoy and Schertzer, 2013). Furthermore, in order to degrade the simulated annual timeseries into irregular paleoclimate-like data we had to use simplified numerical methods to mimic the physical processes and manipulations leading to the recording and retrieval of paleoclimate archives. For our purpose, we chose to low-pass filter the data and sub-sample since this is very common in sedimentary data and ice cores because of, respectively, bio-turbation and diffusion (Dolman and Laepple, 2018; Dolman et al., 2020; Kunz et al., 2020). However, reality is more complex and





other processes could significantly alter the recorded signal and bias the variability estimates in ways not accounted for by our surrogate model. For specific case studies, it is advisable to develop forward models (Stevenson et al., 2013; Dee et al., 2017; Dolman and Laepple, 2018; Casado et al., 2020) of the studied paleoclimate archives in order to understand potential biases generated by the recording and specific sampling.

In summary, it is difficult to ascertain which method is the "best" since the answer depends on the irregularity, the resolution and, most importantly, the underlying $H$ values. In this respect, the HSF is possibly the safer method for paleoclimate applications since it only performs poorly for series with $H < -0.5$, an almost unseen behaviour in climatic timeseries at timescales longer than decadal. Although the LSP gives equally good result for $H \lesssim 0$ and even better near $H = -0.5$, i.e. white noise, its increasing bias and standard deviation for higher $H$ values makes it an uncertain choice for timescales longer than centennial

since climate timeseries at those timescales often show $H \approx 0$ and greater. The MTM produces good estimates for relatively regular series, but struggles with highly irregular timeseries since it must rely on interpolation. While this produces a large bias on the shorter timescales which we have thus discarded, the bias is rather consistent and methods could be developed to correct for it (e.g. Fig. S7). On the other hand, the longer timescales of the MTM are relatively well estimated and it is appropriate to estimate the absolute variance over a timescale band well above the mean resolution (Rehfeld et al., 2018; Hébert et al., 2021).

Further, the effect of proxy biases are well studied for the power spectrum (Kunz et al., 2020; Dolman et al., 2020) mainly due to the known properties of the Fourier transform. Our study is not comprehensive and other methods such as the bias corrected version of Lomb-Scargle (Schulz and Mudelsee, 2002, REDFIT) and the z-transform wavelet (Zhu et al., 2019) exist. More precise estimates of the scaling exponent are obtained by parametric methods based on maximum likelihood, but they require strong assumptions about the underlying process (Del Rio Amador and Lovejoy, 2019). We aimed to cover the most commonly

used methods that also allow for a simple interpretation of variability across timescales, either as they represent the well studied power spectrum or the Haar fluctuations that directly provide changes in amplitude.

## 5 Conclusions

Characterizing the variability across timescales is important to understand the underlying dynamics of the Earth system. It remains challenging to do so from paleoclimate archives since they are more than often irregular and traditional methods to

produce timescale-dependent estimates of variability such as the classical periodogram and the multitaper spectrum generally require regular time sampling.

  We have compared those traditional methods using interpolation with interpolation-free methods, namely the Lomb-Scargle periodogram and the first-order Haar structure function. The ability of those methods to produce timescale-dependent estimates of variability when applied to irregular data was evaluated in a comparative framework. The metric we chose to compare

them is the scaling exponent, i.e. the linear slope in log-transformed coordinates, since it summarizes the behaviour of the variability over a given timescale band. Doing so assumes power-law scaling, a behaviour which is often observed in geophysical timeseries at least approximatively (Mandelbrot and Wallis, 1968; Cannon and Mandelbrot, 1984; Pelletier and Turcotte, 1999; Malamud and Turcotte, 1999; Fedi, 2016; Corral and Gonz\`alez, 2019). To evaluate our estimators, we generated frac-



tional noise as surrogate annual timeseries characterized by a given scaling exponent $H$, also known as fractional Gaussian
noise when they are stationary ($H < 0$) and fractional Brownian motion when they are non-stationary ($H > 0$). The annual
timeseries were then degraded to resolutions characteristics of paleoclimate archives for the recent Holocene.

We found that for scaling estimates in irregular timeseries, the interpolation-free methods are to be preferred over the
methods requiring interpolation as they allow for the utilization of the information from shorter timescales without introducing
additional bias. In addition, our results suggest that the Haar structure function is the safer choice of interpolation-free method
since the Lomb-Scargle periodogram is unreliable when the underlying process generating the timeseries is not stationary.
This conclusion was reinforced by the application to the proxy database which indeed showed the Haar structure function to
give more reliable estimates over a wide range of $H$. Given that we cannot know a priori what kind of scaling behaviour is
contained in a paleoclimate timeseries, and that it is also possible that this changes as a function of timescale, it is a desirable
characteristic for the method to handle both stationary and non-stationary cases alike.

*Code and data availability.* The data is available as a supplement to Rehfeld et al. (2018) and the code is publicly available on BitBucket:
https://bitbucket.org/RphlHbrt/rscaling/src/master/.

*Author contributions.* All authors participated in the conceptualization of the research and the methodology. RH developed the software and
visualization, and conducted the formal analysis and investigation. KR and TL provided supervision. RH prepared the original draft and all
authors contributed to the review and editing of the final manuscript.

*Competing interests.* The authors declare no competing interests.

*Acknowledgements.* This study was undertaken by members of CVAS following discussions at several workshops. CVAS is a working group
of the Past Global Changes (PAGES) project, which in turn received support from the Swiss Academy of Sciences and the Chinese Academy
of Sciences. We particularly thank T. Kunz and S. Lovejoy for in depth discussions, as well as T. Graves and C. Franzke for freely sharing
software for generating fractional noise. We also acknowledge discussions with M. Casado, L. del Rio Amador, A. Dolman, I. Kröner and
T. Münch. We thank all original data contributors who made their proxy data available. Funding by the Deutsche Forschungsgemeinschaft
(DFG, German Research Foundation) project no. 395588486 and the German Federal Ministry of Education and Research (BMBF) through
the PalMod project (grant no. 01LP1926C) is acknowledged. This is a contribution to the SPACE and GLACIAL LEGACY ERC projects;
these projects have received funding from the European Research Council (ERC) under the European Union's Horizon 2020 research and
innovation programme (grant agreement no. 716092 and no. 772852).





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
