# Peer review of "Comparing estimation techniques for temporal scaling in paleoclimate timeseries"

_Nonlinear Processes in Geophysics, 2021_

## Author Comment (AC1)

The paper by Hébert et al. "Comparing estimation techniques for timescale-dependent scaling of climate variability in paleoclimatic time series" studies temporal scaling of paleodata using various techniques and simulated data with non-equidistant time step. The paper is well written and can be accepted after a minor revision, according to the following comments.

The title should be changed to "Comparing estimation techniques for temporal scaling in paleoclimatic time series".

Thank you for this suggestion, we gladly accept to shorten the title in this manner.

The following R package could be added as a reference and possibly as a tested tool:

https://www.atmosp.physics.utoronto.ca/people/vyushin/mysoftware.html

Thank you for bringing this package to our attention, we will reference it in the paper and use it in future work dealing with regularly sampled data. We think however that including a test of those tools would broaden the scope of the paper too much.

We add the reference on line 45:

See this approach used in a paleoclimatology context in Huybers and Curry (2006), Laepple and Huybers (2014a, b) and Rehfeld et al. (2018), and see also an implementation of these methods in R for regular climate data, including functions for statistical testing, scaling exponent estimation and trend estimations for different residual models, provided by Vyushin et al. (2009).

It would be useful to add a list of acronyms in the beginning of the paper.

We added the list of acronyms:

| | |
|---|---|
| CPG | Classical Periodogram |
| LSP | Lomb-Scargle Periodogram |
| MTM | Multitaper Spectrum Method |
| HSF | first-order Haar Structure Function |
| DFA | Detrended Fluctuations Analysis |
| LGM | Last Glacial Maximum |
| fGn | fractional Gaussian noise |
| fBm | fractional Brownian motion |

**Table 1.** Table of acronyms used in this paper.

I think the following paper should be added to references: Detecting long-range correlations with detrended fluctuation analysis, JW Kantelhardt, E Koscielny-Bunde, HHA Rego, S Havlin, A Bunde, Physica A: Statistical Mechanics and its Applications 295 (3-4), 441-454 (2001)

The reference was added.

It would be good to define what the authors mean by "quazi-Gaussian" (line 86).

We propose to add the clause in green in the sentence on line 86:

 "In this work, we will focus on the quasi-Gaussian case, i.e. when statistics approximately follow the Gaussian distribution, in order to minimize the number of estimated parameters;..."

In lines 33, 438, 479, the surnames should be corrected.

Corrected

For ensemble averages, instead of symbols < and > it is better to use \langle and \rangle

Corrected

In line 356, it is not clear what database is meant.

To clarify we added after database: "(see section 2.3)"

In line 468, "Nature Climate Change" is typed twice.

Corrected

In line 490, remove &ndash

Corrected

In line 507, change the titles from capital letters

Corrected

---

## Author Comment (AC2)

In this manuscript submitted to as special issue in paleoclimate time series analysis, the authors consider different analysis methods to estimate the H scaling index of paleoclimate series. For this they consider fractional Brownian motion simulations, with adequate modifications in order to introduce an irregular time step, and they systematically check all proposed methods. This work is well explained and the systematic work is convincing. I have some minor comments and one suggestion of significant change.

Major comment:
To understand and appreciate the results I need to see the estimated \hat{H} versus H, for H \in (-1,1). This plot is the most important. If relevant the authors may plot in the same graph the standard deviation. The bias-std plots (Figs 3,4, 5 and 7) are only providing information on the fluctuations of the estimates. The mean value is the more important information for the reader. As I see from Figure 1, it seems that the spectral method may be less accurate when H<0. This plot should help to visualize this property and any other…

We agree with the reviewer that the bias of an estimator is important, which can be depicted as the plot of estimated $\hat{H}$ versus true H. We might not have made the connection clear enough. The bias is currently defined in Sect. **2.2.2**:
$B = < \hat{H} > -H$
Where B is the bias, $\hat{H}$ is the estimator and H is the input "true" H. Therefore, B corresponds to both the deviations from the 1-to-1 line on a $\hat{H}$ vs H plot, and to the x-axis on the bias-std plots we provided, although on the latter, one needs to carefully look at the legend to see the dependence with H.

To provide an easier to understand characterization of the mean behaviour of the bias as a function of H, we thus propose, as suggested, to include the $\hat{H}$ vs H figure in the main text for the irregularity experiment (Figure 3):

[Figure]

,

and also a similar figure for the length experiment (Figure 5) would be added in the supplement, along with the analogous figures for the block average method, which correspond to the results shown on Figures S1 and S3.

The confusion might also have stemmed from a slight mix up on our part regarding Figure 1 which incorrectly displayed results using the Block Average method in order to degrade the annual timeseries to an irregular resolution, rather than the Fitlering+Sub-Sampling. As a result, the displayed result on Figure 1 shows a bias for the regular case which is not reflected on Figures 3,4,5 and 7 as they were produced with the Filtering+Sub-Sampling method.

The reason for this bias is the aliasing of power from frequencies below the Nyquist frequency which is large in the case of blue noise with β<0 (H<-0.5) since the power keeps increasing below the Nyquist frequency. This is in fact one of the main motivations (lines 250-254) for us to show the Filtering+Sub-Sampling method as the main result, instead of the Block Average, since this allows us to identify other (more subtle) effects more easily rather than just the aliasing bias. The negative bias of the spectral method for β<0 when using the Block Average method can be seen in the supplement on Fig. S1, S2, S3.

We thus propose to update Figure 1 using the Filtering+Sub-Sampling method and introduce an analogous corrected figure in the supplement using the Block Average method in order to provide the same intuitive visualization of the mean behaviour for the Block Average results shown in the supplement.

 Minor comments:
- In line 27, replace log-linear by log-log
"Therefore, processes that fulfill this property show a log-linear relationship in the power spectrum (Schuster, 1898; Percival and Walden, 1993) over a given range of timescales."
By this we meant that once the log is taken we see a linear relationship between the log. We agree that the wording log-linear is indeed confusing.
We thus propose to rewrite as:
"Therefore, the power spectrum of such processes will appear linear on a log-log plot over a given range of timescales (Schuster, 1898; Percival and Walden, 1993). "

- In the review lines 46-52, the authors should cite the literature coming from other fields where irregular sampling had importance, such as astrophysics (where the Lomb-Scargle spectrum was introduced) and also fluid mechanics where Laser Doppler Velocimetry produces irregular sampled velocity data; in this domain an important literature is devoted to the question of adequate procedures for estimating Fourier spectra from LDV measurements.
We propose to add the following citations  on lines 46-52:
"Secondly, the estimator can be adjusted for arbitrary sampling times. The so-called Lomb-Scargle Periodogram (LSP; Lomb, 1976; Scargle, 1982; Horne and Baliunas, 1986) was developed in the field of astronomy to identify periodic components in noisy astronomical timeseries with sampling hiatus and was often used to analyze Laser Doppler Velocimetry experiments which produce irregularly sampled velocity data (Benedict et al., 2000; Munteanu et al., 2016; Damaschke et al., 2018) as well as for the detection of biomedical rhythms (Schimmel, 2001). The LSP has sometimes been used in paleoclimatological context (Schulz and Stattegger, 1997; Trauth, 2020), although it may introduce additional bias and variance (Schulz and Stattegger, 1997; Schulz and Mudelsee,  2002; Rehfeld et al., 2011)."

- reference Corral and Gonzalez seems to have a problem of LateX writing - in the reference list I think it is not necessary to systematicaly provide the web address of each paper: the doi was introduced for this. Doi itself is sufficient (without https://doi.org). Example for the first reference: "doi: 10.1038/nclimate1456"

The mistake in Corral and Gonzalez was fixed.
We agree with the comment regarding web addresses. This was done automatically by the copernicus template. We should verify with the typesetting team if this should be removed.

- in the Discussion section it is necessary to mention the limitation of the present work which addresses only the H exponent for scaling processes, and not the intermittency. They should cite works that showed that the climate proxy data have multifractal statistics and discuss the fact that time series with intermittent fluctuations may react differently to the different methods proposed here, and also that an adequate method must be used to extract intermittency parameters (i.e. all the moment function and not only one moment order). The spectral methods cannot do this.
In the method section we already have a statement acknowledging this (line 89-91):
"On the other hand, highly intermittent archives which clearly display multifractality, such as volcanic series \citep{lovejoy_scaling_2016} or dust flux \citep{lovejoy_spiky_2019}, would require the ``intermittency correction'' from the moment scaling function $K(q)$."

We propose to further add in the discussion the following statements (in green following the original text in red copied here for context):

Around line 401:
"Our conclusions however rely on the real data having similar properties to the surrogate data used to validate the methods, and the conclusions may be different if the real data have properties different than the assumed power-law scaling. It is difficult to say for example which method would perform best if the data analyzed would contain two different scaling regimes, one with low H and the other with high H (Lovejoy and Schertzer, 2013). Another limitation of our surrogate data is that it is Gaussian-distributed, whereas real paleoclimate data can exhibit multifractality (Schmitt et al., 1995; Shao and Ditlevsen, 2016)."

Around line 425:
"On the other hand, the longer timescales of the MTM are relatively well estimated and it is appropriate to estimate the absolute variance over a timescale band well above the mean resolution (Rehfeld et al., 2018; Hébert et al, 2021). Further, the effect of proxy biases are well studied for the power spectrum (Kunz et al, 2020; Dolman et al., 2020) mainly due to the known properties of the Fourier transform. However, the MTM, and the LSP, do not allow for the characterization of intermittency through the study of all statistical moments, contrary to the HSF which is thus better suited for the analysis of timeseries displaying multifractality, such as paleoclimate timeseries at glacial-interglacial timescales comprising Dansgaard–Oeschger events (Schmitt et al., 1995; Shao and Ditlevsen, 2016)"

---

## Author Response (AR2)

Dear Daniel,

Thank you for your swift answer. We have made the corrections as suggested and submitted the revised manuscript. We have summarized below the additions we made in response to the corrections you suggested.

Best wishes,
Raphaël on behalf of the authors

**Editor Decision: Publish subject to technical corrections** (24 May 2021) by Daniel Schertzer
Comments to the Author:
Dear Raphael,

I am pleased to conclude the discussion of your paper "Comparing estimation techniques for temporal scaling in paleoclimate time series" with the decision that the revised version should be published with (possible) technical corrections.
For instance, as you know it, "K(q) is zero for Gaussian processes" (line 80) is rather restrictive since it is only a particular case of uni/mono-fractal processes, although this is one of the most celebrated and studied cases. This also applies to the few places where the term "quasi-Gaussian" appears.

We clarified this by saying that "K(q) is zero for some monofractal processes such as the well-studied Gaussian case, and thus, for these specific cases, all statistical moments scale similarly." on line 80-82.
We also rewrote two instances of "the quasi-Gaussian case" as "the (monofractal) quasi-Gaussian case" on line 91 and line 180.

In relation with your exchange with referee#2 on the importance of intermittency in paleoclimate data, it would be fair to cite Schmitt et al (1995) the first time you mention this problem (line 95), not only in the Discussion section.

We added the reference there and mentioned the multifractality of timeseries on glacial-interglacial timescales. Line 95-97 of the revised manuscript.

I also noted a common oversimplification in Eq.4, because there is in fact a proportionality constant between the power spectrum and the squared Fourier transform (in fact a Dirac function at infinite resolution, the number of points if finite).

This has been corrected by dividing by the temporal coverage of the series. We also made a small correction in the periodogram estimate, multiplying by the resolution to ensure the non-dimensionality of the exponential argument.

Sincerely Yours,

Daniel Schertzer (NPG Executive Editor)